# Neutrophils versus Protozoan Parasites: *Plasmodium*, *Trichomonas*, *Leishmania*, *Trypanosoma*, and *Entameoba*

**DOI:** 10.3390/microorganisms12040827

**Published:** 2024-04-19

**Authors:** Eileen Uribe-Querol, Carlos Rosales

**Affiliations:** 1Laboratorio de Biología del Desarrollo, División de Estudios de Posgrado e Investigación, Facultad de Odontología, Universidad Nacional Autónoma de México, Mexico City 04510, Mexico; 2Departamento de Inmunología, Instituto de Investigaciones Biomédicas, Universidad Nacional Autónoma de México, Mexico City 04510, Mexico

**Keywords:** neutrophil, protozoan, *Plasmodium*, *Trichomonas*, *Leishmania*, Chagas disease, *Trypanosoma*, amoeba, inflammation

## Abstract

Neutrophils are the most abundant polymorphonuclear granular leukocytes in human blood and are an essential part of the innate immune system. Neutrophils are efficient cells that eliminate pathogenic bacteria and fungi, but their role in dealing with protozoan parasitic infections remains controversial. At sites of protozoan parasite infections, a large number of infiltrating neutrophils is observed, suggesting that neutrophils are important cells for controlling the infection. Yet, in most cases, there is also a strong inflammatory response that can provoke tissue damage. Diseases like malaria, trichomoniasis, leishmaniasis, Chagas disease, and amoebiasis affect millions of people globally. In this review, we summarize these protozoan diseases and describe the novel view on how neutrophils are involved in protection from these parasites. Also, we present recent evidence that neutrophils play a double role in these infections participating both in control of the parasite and in the pathogenesis of the disease.

## 1. Introduction

Adaptive immune responses are very specific for particular foreign agents (antigens) and usually long-lasting, but they also take a long time (around two weeks) to develop [1]. Since we are in continuous contact with unicellular microorganisms, the defense of our organism depends to a large extent on the innate immune system [1]. An essential part of the innate immune system is the neutrophils, which are polymorphonuclear and granular leukocytes [2]. Neutrophils are the most abundant (50–70%) circulating leukocytes in human blood. They measure around 7–10 μm in diameter, present a lobulated nucleus, and have many granules and secretory vesicles in their cytoplasm. Neutrophils are quickly recruited in large numbers from the blood into sites of infection or inflammation. For this reason, they are usually the first cells to interact with invading microorganisms and are thus considered a first line of defense of the innate immune system [3]. Neutrophils are recruited to affected sites by chemoattractants, a chemically diverse group of molecules that stimulate the migration of leukocytes and provide guidance to the cells. Chemoattractants include lipids, such as PAF (platelet-activating factor) or LTB4 (leukotriene-B4); N-formylated peptides; complement anaphylatoxins, such as C3a and C5a; and chemokines (host small proteins) [4]. In humans, at least seven chemokines (CXCL1, CXCL2, CXCL3, CXCL5, CXCL6, CXCL7, and CXCL8) have been identified to mediate the recruiting of neutrophils in a timely and coordinated manner [5]. The chemokine CXCL8, also known as interleukin (IL)-8, is the most potent chemoattractant for neutrophils [6]. In an infected tissue, neutrophils carry out important antimicrobial functions [7], including degranulation [8], production of reactive oxygen species (ROS) [9,10], phagocytosis [11], formation of neutrophil extracellular traps (NETs) [12], and trogocytosis [13]. Neutrophils in blood, that do not migrate to infected tissues, turn into aged neutrophils and migrate back to the liver, spleen, or bone marrow [14], where they will die by apoptosis and finally be eliminated by macrophages in a process known as efferocytosis [15].

Neutrophils contain preformed effector molecules stored in their intracellular granules. There are four types of granules in neutrophils: primary or azurophil (containing myeloperoxidase, elastase, and defensins), secondary or specific (containing lactoferrin, cathelicidin, and metalloproteinases), tertiary or gelatinase (containing gelatinase proteins such as matrix metalloproteinase-9 (MMP-9)), and secretory (containing serum albumin, cytokines, and membrane-bound components, such as adhesion molecules and receptors) [16,17]. At sites of infection, neutrophils achieve a rapid response against microorganisms by degranulation, that is, the release of granule proteins toward pathogens to induce their killing and digestion [18]. Another mechanism for killing microorganisms is the generation of ROS. To achieve this, neutrophils activate an NADPH (nicotinamide adenine dinucleotide phosphate) oxidase enzyme complex. NADPH oxidase generates large amounts of superoxide, in a process also known as oxidative burst. Superoxide is a precursor of H_2_O_2_ and other forms of ROS with potent antimicrobial activity [18]. Phagocytosis is arguably the most important antimicrobial function of neutrophils [11,19]. Phagocytosis is a receptor-mediated process that results in internalization of a particle larger than 0.5 μm into the cell. Neutrophils recognize microorganisms through PAMPs (pathogen-associated molecular patterns) or through opsonins (antibody molecules or complement components) to initiate phagocytosis, which results in ingestion of the microorganism into a vacuole called the phagosome. Next, the phagosome fuses with lysosomes, in a process known as phagosome maturation, to becomes a phagolysosome [20]. The interior of the phagolysosome has an acidic pH and many degradative enzymes that are toxic for the microorganism ingested [18]. Neutrophils can also form and release NETs through a dynamic cell death program known as NETosis. NETs are fibers of decondensed chromatin decorated with histones and antimicrobial proteins and enzymes from the neutrophil granules. Once NETs are deployed, they function as a physical barrier where pathogens get trapped and could be eliminated extracellularly and independently of phagocytosis [18]. NETs are beneficial when controlling the dissemination of infectious microorganisms. However, unregulated NET formation can also be detrimental to the host, since excessive NETs may promote inflammation and tissue damage [12]. Trogocytosis (from the Greek trogo-: nibble) is a recently identified cellular process by which one cell physically takes little pieces (“bites”) from another cell and ingests these pieces of cellular material. Through trogocytosis, a cell can damage the membrane of another cell, leading to its death [13]. Neutrophils can implement trogocytosis to kill large cells, such as sperm cells and some tumor cells [21,22]. In addition to these important antimicrobial functions, in recent years, it has become evident that neutrophils are also key effector cells of the adaptive immune system [18], and display phenotypic heterogeneity and functional versatility [23,24].

In a classical view, neutrophils are reported to be efficient cells to eliminate pathogenic bacteria and fungi [25]. However, much less is known about the role of neutrophils in infections caused by protozoan parasites. This is somewhat surprising, since several protozoan parasites are responsible for important diseases around the world, for example malaria, leishmaniasis, and amoebiasis. These diseases affect millions of people globally and represent a serious burden to human health, as indicated by the 2013 Global Burden of Disease Study [26,27]. In most cases of protozoan parasite infections, a large number of infiltrating neutrophils are observed in the affected tissues [28,29,30,31,32]. This suggests that neutrophils are relevant immune cells for controlling the infection. Yet, in most cases, there is also a strong inflammatory response that can provoke tissue damage. Therefore, the role of neutrophils in protozoan parasitic infections remains controversial. In this review, we summarize the protozoan diseases malaria, trichomoniasis, leishmaniasis, Chagas disease, and amoebiasis, and describe the novel view of how neutrophils are involved in protection from these parasites. Also, we present recent evidence that neutrophils play a double role in these infections, participating both in control of the parasite and in the pathogenesis of the disease.

## 2. Malaria

Malaria is a disease caused by protozoan parasites from the genus *Plasmodium*. There are five species within the genus *Plasmodium* recognized to infect humans: *P. falciparum*, *P. vivax*, *P. malariae*, *P. ovale*, and *P. knowlesi* [33]. According to the World Health Organization (WHO), malaria is a serious disease globally producing a huge public health problem with more than 241 million clinical cases and 627,000 deaths just in 2020 [34]. The disease causes severe morbidity and mortality in African countries south of the Sahara Desert, where more than 90% of all malaria cases and deaths are reported [27]. Young children and pregnant women are especially susceptible to this disease, which in most cases is due to infections by *P. falciparum* [34]. Clinically, malaria is found in three forms: asymptomatic, mild or uncomplicated, and severe or complicated [35,36]. The asymptomatic form is found in the vast majority of infected individuals. When symptoms develop, in the mild form, they include fever and sweating, chills, fatigue, nausea or vomiting. These symptoms are thought to result from a combination of the sequestration of infected erythrocytes in the microvasculature, activation of endothelial cells, and pro-inflammatory and pro-coagulant responses. In the severe form, *Plasmodium* infection becomes complicated by abnormalities in the patient’s blood or metabolism or by serious organ failures. The manifestations of severe malaria include: cerebral malaria, characterized by neurologic alterations, including abnormal behavior, impairment of consciousness, seizures, or coma. Also, severe anemia, due to destruction of erythrocytes, hemoglobinuria, and even acute respiratory distress syndrome (ARDS), an inflammatory reaction in the lungs that inhibits oxygen exchange, may also be observed. In severe cases, complications can lead to death [37].

*Plasmodium* protozoans are complex organisms that are obligate parasites of vertebrates and insects. The life cycle of *Plasmodium* species involves two phases. In one phase, the parasite develops in a blood-feeding female *Anopheles* mosquito host. In another phase, the mosquito injects parasites into a vertebrate (human) host during a blood meal [38]. The bite of the female *Anopheles* mosquito introduces *Plasmodium* sporozoites into the human host. Sporozoites mature as they travel to the liver and ultimately the bloodstream. Initially, during a blood meal from an infected person, *Plasmodium* gametocytes enter the midgut of the mosquito where they transform into male microgametes and female macrogametes. The union of gametes forms a zygote, which transforms into an ookinete that penetrates the intestinal wall of the mosquito. Then, the ookinete is converted into an oocyst. Inside the oocyst, sporozoites develop, which then migrate to the mosquito salivary gland. The sporozoites are released during the blood meal of the mosquito, entering the vertebrate host. Inside a human, sporozoites migrate and infect the liver, where they undergo a single round of replication becoming merozoites. Next, merozoites exit the liver and enter the bloodstream where they infect erythrocytes. *Plasmodium* merozoites continue dividing inside erythrocytes, causing their destruction and releasing more merozoites. The parasites then go through continuous cycles of erythrocyte infection. A small number of parasites differentiate into a sexual stage called a gametocyte, which is picked up by a mosquito during a blood meal, and in this way completing the life cycle [39]. The continuous destruction of erythrocytes is the main cause of malaria.

Although *Plasmodium* merozoites, the causal agents of malaria, are found in the blood stream, it is surprising that there are relatively few studies exploring the role of neutrophils in malaria. This may be in part due to technical difficulties, including the short lifespan of neutrophils and the complications involved in isolating these leukocytes from blood, which make fieldwork a real challenge in malaria-endemic countries [40]. However, recent evidence indicates that neutrophils play a double role in malaria, participating both in control of the parasite and in the pathogenesis of the disease [41]. During an infection with *Plasmodium*, it has been estimated that blood neutrophil counts can increase up to 40%, and the number of leukocytes is associated with parasitemia [42]. However, neutrophil counts did not change with the severity of the disease [43]. Thus, it seems that different neutrophil defense mechanisms may influence the outcome between protection and pathogenesis in malaria [41].

### 2.1. Neutrophil Response Anti-Plasmodium

#### 2.1.1. Phagocytosis

Neutrophils are normally the first leukocytes to respond to an invading pathogen, and phagocytosis is a common defense mechanism of these cells. Phagocytosis of free merozoites or gametocytes has been observed in vitro [44] and in blood smears from patients with malaria [45]. This phagocytosis seems to be dependent on complement activation [45]. However, it is still uncertain to what extent phagocytosis of free parasites controls *Plasmodium* proliferation [46]. Phagocytosis of infected erythrocytes, which is observed more frequently in malaria patients, may be more relevant to control of *Plasmodium* growth. This phagocytosis is mainly dependent on the presence of antibodies [47] and apparently independent of complement [48]. Therefore, this response may not be relevant during primary infections, since it depends on previous and chronic exposure to parasites, but it may become important in controlling parasite burden in malaria-endemic areas [49]. Although phagocytosis of *Plasmodium*-infected erythrocytes is easily detected in vitro, phagocytosis in vivo is usually inferred by the presence of hemozoin (malaria pigment) within neutrophils [50]. The number of neutrophils with malaria pigment in peripheral blood increases with disease severity [51,52] and correlates with parasitemia and mortality due to severe malaria in adults and children [53]. Hemozoin is the end product of hemoglobin digestion by parasites, and it is able to inhibit further phagocytosis of infected erythrocytes by phagocytes that previously ingested hemozoin [54,55]. Hence, neutrophil phagocytosis is capable of controlling *Plasmodium* infections to a certain extent, by ingesting free parasites and, more importantly, infected erythrocytes (Figure 1).

#### 2.1.2. Reactive Oxygen Species (ROS)

In vitro experiments suggest that during *Plasmodium* infection, neutrophils become activated and produce ROS, which may have a role in parasite clearance. Neutrophils from children with malaria inhibit parasite growth better than neutrophils from uninfected children or adults [56], and neutrophils from children with faster parasite clearance times produce more ROS [57]. In addition, antibodies against merozoite antigens, such as PfMSP5 (*Plasmodium falciparum* merozoite surface protein-5) and MSP1 (merozoite surface protein-1), can induce neutrophils to produce more ROS [58,59,60]. Interestingly, antibodies against infected erythrocytes do not seem to induce a stronger ROS production by neutrophils [58]. Therefore, malaria antigens can activate neutrophils to produce ROS, which, in turn, can contribute to destruction of the parasite (Figure 1).

#### 2.1.3. Degranulation

Several neutrophil granule proteins have been associated with protection from infection while studying naturally occurring malaria in Gambian children. Individuals with higher expression of cathepsin G and MMP-9 showed a better inhibition of parasite growth [61]. In vitro experiments indicated that MMP-9 could damage parasites directly, thus acting as a classical antimicrobial protein. In contrast, cathepsin G acted on erythrocytes by cleaving membrane molecules necessary for parasitic invasion [61]. These findings suggest that neutrophils can also control parasite burden by degranulation (Figure 1).

#### 2.1.4. Neutrophil Extracellular Traps (NETs)

During malaria, several factors may induce neutrophils to produce NETs (Figure 1). For example, *Plasmodium*-infected erythrocytes released MIF (macrophage migration inhibitory factor), which in turn induced NET formation [62]. Also, upon rupture of infected erythrocytes, crystal uric acid, and its precursor hypoxanthine, are released [63]. Uric acid crystals are a potent inducer of NET formation [64]. Similarly, erythrocyte rupture releases heme, which can also induce NET formation [65]. In addition, there is some evidence that in vitro neutrophils release NETs in response to *P. falciparum* antigens [66]. Thus, NETs may contribute to sequester parasites and prevent their dissemination. However, malaria pathology is also closely associated with the sequestration of parasites in the microvasculature. Hence, excessive NETs can lead to the development of dense aggregates that promote parasite sequestration in the blood vessels of vital organs, such as liver and lungs. In addition, NET components released by the action of plasma DNase I can also result in higher expression of ICAM-1 (intercellular adhesion molecule-1) on endothelial cells, leading to sequestration of infected erythrocytes in various tissues [65]. Consequently, the interplay between NET formation and parasite sequestration represents a critical aspect of malaria pathogenesis [65]. Thus, inhibition of NETosis may be a therapeutic strategy for vascular complications in malaria.

### 2.2. Anti-Neutrophil Responses in Malaria

Even though neutrophils can eliminate *Plasmodium* parasites through various mechanisms, the mosquito and the parasite express molecules that restrict neutrophil responses and give the parasite an advantage. The mosquito produces in its salivary glands the protein agaphelin, which is increased upon infection with *P. falciparum.* Agaphelin can inhibit neutrophil elastase activity, neutrophil chemotaxis, and NET formation [67]. Also, the antigen-5 salivary proteins, which function as ROS scavengers, can prevent neutrophil destruction of the parasite [68]. In addition, some *Plasmodium* antigens such as histamine-releasing factor and MSP1 can also inhibit neutrophil responses [69]. In a murine malaria model, histamine-releasing factor was found to block neutrophil IL-6 secretion in the liver and then promote parasite development in that organ [70]. In addition, the *P. falciparum* protein MSP1 was found to inhibit neutrophil chemotaxis in vitro by blocking neutrophil responses to proinflammatory protein S100P [71]. Clearly, neutrophils are important cells for controlling *Plasmodium* parasites, but both the mosquito and the parasite itself have evolved mechanisms to evade neutrophil functions and perpetuate infections.

### 2.3. Invasive Bacterial Disease

A frequent complication of malaria, related to neutrophil function, is the increase of bacterial infections disseminating to blood and other organs [72]. A systemic bacterial infection, or sepsis, can lead to a systemic inflammatory response resulting in life-threatening organ dysfunction and death [73]. Because systemic bacterial infections are associated with high mortality rates and with long-term, life-changing sequelae, they remain a global health issue [74]. In developed countries, sepsis is most frequently associated with *Staphylococcus aureus*, particularly methicillin-resistant bacteria, and *Escherichia coli* infections [75,76]. In contrast, in developing countries in Africa, community-acquired bacteremia is often associated with *Salmonella enterica*, habitually nontyphoidal *Salmonella* (NTS) [77]. Invasive NTS is also observed more often among children with *P. falciparum* malaria [72,78]. Additionally, there is evidence that individuals presenting the sickle cell trait (that protects them from malarial anemia) have a lower risk of contracting invasive NTS [79,80]. Hence, there is a clear causal association between malaria and invasive NTS.

Neutrophils are fundamental for elimination of NTS. Bacteria are phagocytosed by neutrophils and then killed in the phagosome by ROS. However, during an acute malaria infection, the capacity of neutrophils to produce an oxidative burst and kill phagocytosed *Salmonella* is greatly diminished [81]. Consequently, resistance to invasive NTS is reduced, and neutrophils become a new niche for these intracellular bacteria to replicate.

The mechanisms triggering inhibition of neutrophil antimicrobial functions are complex and remain unclear. During the *Plasmodium* life cycle within erythrocytes, parasites feed on hemoglobin and store the waste product hemozoin in vesicles denominated as digestive vacuoles (DVs) [82]. Many DVs are also released into the circulation where they can interact with neutrophils and inhibit some of their functions. Neutrophils can phagocytose DVs and initially induce an oxidative burst [54]. However, these cells showed a reduced capacity to kill bacteria due to an impaired subsequent response to produce ROS [54]. Similarly, hemolysis and some hemoglobin products seem to be a major contributor to impaired neutrophil functions. During malaria infection, the parasite continuously ruptures infected erythrocytes. This is also followed by eryptosis of many uninfected erythrocytes [83]. The destruction of erythrocytes results in the release of hemoglobin and its breakdown product heme into the plasma. Elevated levels of heme in plasma were found in both acute [84] and subclinical [85] *P. falciparum* malaria in humans, and in acute *P. yoelii* infection in mice [81,86]. High circulating heme levels resulted in low phagocytosis capacity of *Salmonella* by neutrophils [84]. In addition, in vitro pretreatment of neutrophils with heme also resulted in reduced phagocytosis of *E. coli* [87]. Moreover, during a malaria infection, neutrophils display reduced migration into infected tissues including blood [81], intestine [88], and liver [89]. Neutrophils from *Plasmodium*-infected mice were reported to express heme oxygenase-1(HO-1) [81], an inducible enzyme that degrades heme into iron, carbon monoxide, and biliverdin, and that has been reported to inhibit neutrophil migration into inflamed lungs [90]. Thus, it seems there is a connection between *Plasmodium*-induced HO-1 expression and impairment of neutrophil migration. Additionally, elevated levels of systemic IL-10 are found in highly inflammatory diseases such as sepsis [91,92] and malaria [93,94]. This anti-inflammatory cytokine is well described to affect neutrophil functions, reducing migration to anaphylatoxins, and decreasing bacterial clearance [95,96]. Therefore, multiple mechanisms are involved in inhibition of neutrophil antimicrobial functions, and much more work is needed to fully characterize neutrophil function during malaria.

## 3. Trichomoniasis

Trichomoniasis is a common sexually transmitted infection (STI) caused by the highly motile extracellular flagellated protozoan parasite *Trichomonas vaginalis* [97,98]. It mainly affects individuals with multiple sexual partners and concurrent STIs. Trichomoniasis is most prevalent among women than among men, showing infection rates of 0.5% in men and 1.8% in women [99]. Epidemiological data on trichomoniasis vary globally, with over 143 million new cases reported annually. However, due to underreporting and asymptomatic cases, the true prevalence of trichomoniasis may be underestimated in many regions [100]. In the United States, trichomoniasis is among the most prevalent nonviral STIs, with around 3.7 million infections every year [101]. Prevalence rates can vary significantly among different populations and regions, with higher rates observed in specific demographic groups such as Black or African American individuals and those with lower socioeconomic status or a history of other STIs [101]. Infection of both men and women is, in many instances, asymptomatic. In women, when symptoms develop, they include vaginal discharge, itching, dysuria, and abdominal pain. In men, symptoms are much less frequent and may include prostatitis, decreased sperm motility, and epididymitis [102]. Infections are commonly treated with 5-nitroimidazole drugs such as metronidazole or tinidazole. Unfortunately, antibiotic-resistant *T. vaginalis* strains are on the rise, making treatment of trichomoniasis difficult [103].

As an extracellular parasite, *T. vaginalis* adheres to epithelial cells in the urogenital tract, such as cervicovaginal and prostate epithelial cells, to colonize the human host [104]. In addition, during trichomoniasis, neutrophils are found in large numbers in wet mount smears from vaginal discharges and penile urethral samples [28]. Thus, the parasite interacts with these innate immune system cells [28,29]. From mouse models of trichomoniasis, it is clear that large numbers of neutrophils are recruited quickly to the vagina after inoculation with the parasite [105], and there is evidence that neutrophils are also able to kill these parasites [106,107]. However, neutrophils can also cause tissue damage and enhance inflammatory pathologies [2], and they may therefore be responsible for many symptoms associated with trichomoniasis. Thus, whether neutrophil activity during trichomoniasis is beneficial or detrimental to the host remains unclear.

### 3.1. Neutrophil Response Anti T. vaginalis Parasites

#### 3.1.1. Neutrophil Migration towards *T. vaginalis* Parasites

Epithelial cells at the infection site produce LTB4, an eicosanoid lipid mediator that promotes extravasation of neutrophils [108]. Once in the tissues, neutrophils follow chemotactic signals to home directly to parasites. Because *T. vaginalis* parasites also produce LTB4 [109], neutrophils follow this cue towards the parasites. In addition, neutrophils themselves produce LTB4, continuing a positive feedback loop that enrolls more neutrophils towards the parasites [110]. In response to LTB4, neutrophils display a swarming behavior around pathogens [111] (Figure 2). In the case of *T. vaginalis*, neutrophil swarming around parasites has been observed on vaginal smears [112] and on in vitro co-cultures of neutrophils isolated from blood and axenically grown trichomonads [113]. Additionally, in response to a *T. vaginalis* infection, neutrophils [114] and other immune cells [115] also secrete IL-8. Consequently, large numbers of neutrophils are found in tissues of *T. vaginalis* infection, presumably recruited to control the parasites. However, it has also been reported that higher levels of LTB4 [116] and IL-8 [115] in infected patients correlate with more severe symptoms, suggesting that neutrophils also contribute to pathogenesis of trichomoniasis.

#### 3.1.2. Neutrophils Kill *T. vaginalis* by Trogocytosis

The classical mechanisms used by neutrophil to eliminate microorganisms include phagocytosis, degranulation of antimicrobial molecules, and the formation of NETs [24,117]. To determine what mechanism was used by neutrophils to kill *T. vaginalis* parasites, each of the neutrophil antimicrobial mechanisms were systematically inhibited. Surprisingly, none of the classical antimicrobial mechanisms were found to be responsible for killing these parasites. Still, neutrophils rapidly killed *T. vaginalis* in a dose-dependent and contact-dependent manner [113] The process involved neutrophils surrounding the parasite and taking small pieces of the parasite membrane [113] in a novel mechanism known as trogocytosis [13]. Hence, neutrophils swarm around *T. vaginalis* parasites and kill them by damaging their membrane through trogocytosis (Figure 2). Intriguingly, neutrophils only performed trogocytosis on live *T. vaginalis*, while dead parasites were eliminated by phagocytosis [113]. This activity is reminiscent of the amoeba *Entamoeba histolytica*, which nibbles live intestinal cells and carries out phagocytosis on dead epithelial cells [118,119].

### 3.2. Neutrophil Evasion Tactics of T. vaginalis

In spite of the capability of neutrophils to kill *T. vaginalis*, many infected patients cannot clear the parasites on their own, and antibiotic therapy is necessary to control the infection. This suggests that *T. vaginalis* uses evasion tactics to prevent killing from neutrophils. Several anti-neutrophil tactics of *T. vaginalis* are described next.

We mentioned that neutrophils migrate towards parasites to cover them and perform trogocytosis. However, it was also reported that trichomonads actively move away from activated neutrophils. In transwell experiments, parasites displayed reduced migration towards neutrophils on the other side of the filter. This effect was blocked when activated neutrophils were treated with catalasa or superoxide dismutase to break down ROS [107]. Thus, *T. vaginalis* parasites were vigorously kept away from ROS produced by activated neutrophils [107]. Still, this chemorepulsion process was observed at longer times (45 min) than trogocytosis (usually within 15 min). So, it is not clear how chemorepulsion would help parasites to avoid killing by trogocytosis. However, in tissues, the process of chemorepulsion may help parasites avoiding zones where active neutrophils could trap them.

*T. vaginalis* parasites also tend to form aggregates that promote epithelial cell destruction and induce pathogenesis [120,121]. In fact, more pathogenic strains of *T. vaginalis* form more clusters in vitro [122,123], suggesting that aggregation is a virulence factor of this parasite. Within a cluster, it would be difficult for several neutrophils to cover individual parasites, thus making trogocytosis inefficient. Also, neutrophils would first trogocytose parasites on the outside of the cluster, giving the trichomonads inside a better chance to survive.

Neutrophils are short-lived cells that normally undergo apoptosis at the end of their lifetime [124]. Yet, in the presence of *T. vaginalis* parasites, neutrophil apoptosis was significantly higher [125,126]. To confirm that apoptosis was the mechanism killing neutrophils when cultured with *T. vaginalis* during 12 h, neutrophils and parasites were cultured together in the presence of a caspase-3 inhibitor. This resulted in inhibition of apoptosis [125]. In addition, since ROS are potent inducers of apoptosis in neutrophils [127], neutrophils were pretreated with an inhibitor of NADPH oxidase, diphenyleneiodonium chloride (DPI). This also resulted in inhibition of apoptosis [126]. Together, these reports suggest that apoptosis of human neutrophils induced by *T. vaginalis* involves a ROS-dependent activation of caspase-3. However, the apoptosis process is much longer than the time required for neutrophils to kill parasites via trogocytosis. Thus, it seems unlikely that individual parasites could evade killing by inducing neutrophil apoptosis. Still, this mechanism may be relevant in perpetuating an infection.

Large numbers of neutrophils are found in tissues infected with *T. vaginalis*, where they can kill parasites by trogocytosis. However, in infected patients, higher levels of neutrophil chemoattractants also correlate with more severe disease, probably due to neutrophil-mediated inflammation and tissue damage. Therefore, whether neutrophil activity during trichomoniasis is beneficial or detrimental to the host remains unclear.

## 4. Leishmaniases

Leishmaniases are a group of diseases caused by protozoan parasites of the genus *Leishmania*. More than twenty different species of *Leishmania* are capable of infecting humans and causing the various forms of leishmaniasis. The main clinical forms of leishmaniasis are cutaneous, mucocutaneous, and visceral (also known as kala-azar) [128,129]. The cutaneous form is the most common and usually self-healing, while the mucocutaneous form is the most disabling because it appears on skin and mucosal tissues of the nose and mouth. The visceral form develops in spleen and liver and is the most severe form, usually resulting in death [129]. More than 80 countries in America, Africa, and Asia are endemic areas for leishmaniasis, with an estimated more than one million new cases of cutaneous leishmaniasis and about 30,000 new cases of visceral leishmaniasis each year [130]. The severity of the disease varies among other factors with the *Leishmania* species involved. For example, cutaneous lesions caused by *L. mexicana* or by *L. major* are resolved within three months. In contrast, lesions caused by *L. braziliensis* take longer to be resolved [131]. Similarly, mucocutaneous leishmaniasis is endemic in Latin America and is caused mainly by *L. braziliensis*, *L. panamensis*, and *L. amazonensis* [131,132]. Visceral leishmaniasis, in Africa and Asia, is caused mainly by *L. donovani* complex (including *L. donovani donovani* and *L. donovani infantum*), and in America by *L. infantum chagasi* [133].

*Leishmania* protozoa are obligated intracellular parasites. They exist in two stages during their life cycle: a flagellated promastigote that lives in the midgut of an infected female sandfly and an amastigote that lives within cells of a vertebrate host. *Leishmania* promastigotes are transmitted to humans when a female sandfly of *Phlebotomus* (in Africa and Asia) or *Lutzomia* (in New World) genera takes a bloodmeal. Promastigotes are deposited in the skin epidermis or the upper layer of the dermis, where they are rapidly internalized by immune cells, mainly neutrophils and macrophages. Within the cell, promastigotes transform into amastigotes, which rapidly replicate and disseminate to other cells causing disease [134]. The life cycle of the parasite is completed when a sandfly takes a bloodmeal with parasitized cells. In the midgut of the sandfly, amastigotes quickly transform back into promastigotes.

During a bloodmeal, a sandfly causes skin damage and, as a result, inflammation is generated, leading to recruitment of immune cells, predominantly neutrophils. Skin keratinocytes sense the presence of promastigotes through innate immune receptors. Particularly, Toll-like receptor (TLR) 2 detects *Leishmania* phosphoglycans and activates these cells to release CXCL1, CXCL2, and CXCL5 chemokines for neutrophil recruitment [135,136]. In addition, some factors in the sandfly saliva, such as the sand fly salivary yellow proteins (~45 kDa) [137], and the promastigote secretory gel [138] may also contribute to neutrophil recruitment [136]. The promastigote secretory gel is made of proteophosphoglycans secreted from *Leishmania* in the sand fly midgut. The gel forms a plug in the insect gut to facilitate the regurgitation of infective parasites [138]. Thus, neutrophils are the first innate immune cells that interact with *Leishmania* promastigotes and have an important part in phagocytosis and destruction of these parasites [139] (Figure 3). However, since *Leishmania* are intracellular parasites, they have also evolved mechanisms to avoid elimination by neutrophils [140].

### 4.1. Dual Role of Neutrophils in Leishmaniasis

The cellular mechanisms related to *Leishmania* infection are only partially known, and most of our current understanding comes from experimental models of leishmaniasis. Based on these models, it is clear that the early and abundant presence of neutrophils at the site of *Leishmania* inoculation is not sufficient to control this parasite. Instead, it seems that neutrophils help in spreading the infection to other cells. Initially, neutrophils recognize the promastigote lipophosphoglycan via receptors, such as TLR2 and TLR4 [135,141], or complement receptors, such as CR3 [142]. This interaction induces neutrophil activation, resulting in degranulation, phagocytosis, and LTB4 production [143,144]. LTB4, in turn, causes neutrophil swarming [145] and more accumulation of cells around the parasite. In addition, neutrophils produce cytokines, particularly IL-1β, TNF (tumor necrosis factor)-α, TGF (tumor growth factor)-β, and IL-6, that stimulate recruitment of macrophages and activation of other immune cells [139,146].

Neutrophils quickly phagocytose many promastigotes [147] (Figure 3). However, destruction of promastigotes by neutrophil phagocytosis varies among *Leishmania* species. For example, *L. braziliensis* [148] and *L. donovani* [149] are susceptible to degradation within neutrophils. In contrast, *L. amazonensis* [150] and *L. mexicana* [151] can survive within neutrophils. But, in the case of *L. amazonensis*, killing seems to depend on the stage of the parasite. Promastigotes were killed while amastigotes could survive within neutrophils [150]. The mechanism for this difference is not known. However, it might be related to the cytokines induced by each form of the parasite. Promastigotes trigger more TNF-α secretion, and by 18 h, more than 65% of promastigotes were killed by neutrophils. In contrast, amastigotes induced secretion of anti-inflammatory IL-10, and by 18 h, most amastigotes were still alive [150]. These reports clearly show that phagocytosis efficiency of neutrophils is different for the various *Leishmania* species. In fact, because both promastigotes and amastigotes, depending on the species, can survive in neutrophils, they use these leukocytes transitorily to finally gain access to macrophages, their final host cell. Parasites may be released by dying neutrophils and be taken up by macrophages. However, macrophages more frequently take up infected apoptotic neutrophils by a process known as efferocytosis [146]. In this way, parasites can infect macrophages unnoticed. The process has been described as neutrophils being a “Trojan horse” into macrophages [152] (Figure 3). Hence, by resisting the antimicrobial activity and stimulating the apoptosis of neutrophils, *Leishmania* enhances parasite spread into other phagocytic cells, such as macrophages and dendritic cells [153,154]. Therefore, neutrophils play a dual role, preventing or promoting leishmaniasis. First, shortly after promastigote infection, neutrophils reduce parasite numbers, but later neutrophils facilitate safe passage of surviving parasites to other host cells [29,155].

### 4.2. Neutrophil Response Anti-Leishmania

#### 4.2.1. Phagocytosis and Degranulation

Phagocytosis is arguably the most important antimicrobial function of neutrophils [11]. Microorganisms are taken up inside a new vesicle, the phagosome, which undergoes a maturation process by fusing with other neutrophil vesicles and granules to become a phagolysosome [20,156]. Inside the phagolysosome, an acidic, oxidative, and degradative environment is formed. Most microorganisms cannot survive in this hostile environment. For this reason, several microorganisms have evolved various mechanisms to avoid phagocytosis or to escape from phagolysosomes [157]. In contrast, *Leishmania* parasites do not avoid phagocytosis. Instead, they have evolved various strategies that allow them to survive and even replicate within phagolysosomes.

Despite the importance of neutrophil phagocytosis in the early stages of *Leishmania* infections [140,151], little is known about phagolysosome formation in neutrophils after phagocytosis of promastigotes. Most of our current knowledge comes from studies of phagocytosis of *Leishmania* by macrophages. After being phagocytosed, promastigotes must transform into amastigotes to continue the infection process. An initial strategy for survival is to allow time for the parasite to complete this transformation. Some *Leishmania* species, such as *L. donovani* and *L. major*, can interrupt phagolysosome maturation by inhibiting acidification [158], the assembly of the NADPH oxidase for production of ROS [159], or the fusion of the phagosome with endosomes [160]. In contrast, *L. amazonensis* does not seem to restrict phagolysosome maturation. Phagosomes containing parasites become acidic and display lysosomal markers, such as the GTPase Rab7 and LAMP1 (lysosomal associated membrane protein 1), on their membranes within 30 min post infection [161,162]. This process is even faster when infection begins with amastigotes [161]. Thus, *Leishmania* amastigotes are clearly more resistant to lysosomal enzymes. In addition, the final parasite-containing phagosome is not a mature phagolysosome, but rather a hybrid vesicle that displays molecules associated with the endoplasmic reticulum. This hybrid vesicle is called the parasitophorous vacuole [163]. Moreover, the organization of the parasitophorous vacuole varies among different *Leishmania* species. For *L. major* and *L. donovani*, parasitophorous vacuoles are tight and contain a single amastigote, while for *L. mexicana* and *L. amazonensis*, the vacuoles are large and contain several amastigotes, usually connected to the vacuole membrane [162].

In the case of neutrophils, parasitophorous vacuoles seem to be formed mainly with the granules present in these cells. In particular, fusion of parasite-containing vesicles with myeloperoxidase-containing azurophilic granules has been reported for human neutrophils infected in vitro with *L. major* and *L. donovani* [164]. Also, parasitophorous vacuoles in neutrophils are less acidic and capable of producing much more ROS than the vacuoles in macrophages. These differences are due to lower expression of V-ATPase molecules (which pump proton ions into the phagosome) and higher activity of NADPH oxidase [165,166]. Despite the high production of ROS, this antimicrobial mechanism does not appear to cause damage to the parasites. In vitro infection of neutrophils with *L. amazonensis* results in efficient ROS production, but this does not seem to significantly affect the survival of parasites nor the development of disease [150,167]. Instead, ROS seem to be required for proper induction of neutrophil apoptosis, which is relevant for perpetuating the infection, particularly during chronic disease [168]. Clearly, *Leishmania* parasites, particularly amastigotes, are resistant to antimicrobial functions of neutrophils and are even capable of replicating within these leukocytes [151]. Still, much is unknown about the biology of parasitophorous vacuoles in neutrophils. Particular interesting questions are how the different species of Leishmania resist the various antimicrobial mechanisms of neutrophils, as well as how differences in phagolysosome maturation affect the promastigote to amastigote transition for each Leishmania species, and what consequences these differences have on disease.

#### 4.2.2. Neutrophil Extracellular Traps (NETs)

NETs have been found in biopsies of cutaneous leishmaniasis lesions and in vitro can be induced by promastigotes or amastigotes of various *Leishmania* species, including *L. major*, *L. mexicana*, *L. amazonensis*, and *L. infantum* [140,169] (Figure 3). Also, the purified lipophosphoglycan of *L. amazonensis* is capable of inducing NET formation [169,170]. Promastigotes seem to be more efficient than amastigotes at inducing NETosis [169,171], but both parasite forms are susceptible to the toxic activity of histones, in a process that is also dependent on neutrophil elastase activity [170]. Opposing this, *Leishmania* parasites can resist the effect of NETs through increased expression of the surface protein gp63 (a zinc-metalloproteinase), making them less susceptible to histone H2A [170], and also by reducing expression of the lipophosphoglycan protein. In addition, the parasite enzyme 3′nucleotidase/nuclease, which can degrade NETs, allows the release of promastigotes from NETs [172].

A critical unresolved issue about NETosis is how neutrophils “decide” between performing phagocytosis or forming NETs. Our current interpretation is that the two functions are mechanistically irreversible and mutually exclusive [173]. In the case of *Leishmania*, NETs can kill the parasites. However, phagocytosis, which is advantageous for perpetuating infection, is observed more frequently.

## 5. Chagas Disease

Chagas disease, also known as American trypanosomiasis, is a complicated serious and potentially life-threatening illness caused by the protozoan parasite *Trypanosoma cruzi* [174,175]. Chagas disease is the most important parasitic disease in Latin America, concentrating in endemic areas of many countries. However, it is estimated that 6–7 million people are infected worldwide [176,177]. The WHO has included Chagas disease among the 20 neglected tropical diseases, since 28,000 new infections and 14,000–50,000 deaths occur every year, and 70–100 million people are at risk of infection [178]. Chagas disease was discovered by the Brazilian medical researcher Carlos Chagas in 1909, who also identified the etiological agent *T. cruzi* [177,179]. This parasite is mostly transmitted to humans by contact with feces of blood-feeding triatomine insects known as “kissing bugs” [177]. *Triatoma infestans*, *Rhodnius prolixus*, and *Triatoma dimidiata* are the only competent insect vectors capable of transmitting *T. cruzi* to humans. *T. infestans* is found mainly in sub-Amazonian endemic regions, *R. prolixus* is found in South and Central America, and *T. dimidiata* is found in Mexico [180]. Chagas disease also becomes a complex disease due to variation among insect vectors, which results in numerous ways to infest homes, become resistant to insecticides, and transmit *T. cruzi* [181,182]; and also due to a wide genetic diversity in the parasite itself. At least six genetic lineages or DTUs (discrete typing units) of *T. cruzi* have been identified: TcI− TcVI [183,184].

Chagas disease presents two clinical phases: acute and chronic. In the acute phase, high parasitemia is present, often accompanied by systemic symptoms, such as fever, headache, and diarrhea [175]. Afterwards, most infected people continue with an asymptomatic phase, in which infection persists undetected while the parasite slowly replicates in tissues. However, around 30% of infected people advance to the chronic phase 10–30 years after the initial infection. In the chronic phase, several organs are affected by a strong inflammatory response leading to cardiac, digestive, or neurological alterations, which can lead to death [174]. Particularly, Chagas cardiomyopathy is the main cause of fatality [185]. Despite the fact that Chagas disease is also classified by the WHO as the most prevalent of the poverty-caused and poverty-promoting neglected tropical diseases [186], clinical treatment involves only two drugs: nifurtimox (developed in 1960) and benznidazole (developed in 1972). Both drugs have low cure rates and, in addition, present side effects that may result in the interruption of the treatment. Thus, new drugs against *Trypanosoma* are urgently needed [187].

The life cycle of *T. cruzi* involves two hosts and four parasite stages. The infection of a mammalian host begins with non-replicative flagellated metacyclic trypomastigotes present in feces from a triatomine insect. Metacyclic trypomastigotes penetrate the skin through the insect bite wound. They can also enter via several mucosal membranes. Once in tissues, metacyclic trypomastigotes can infect many types of nucleated cells, entering into a parasitophorous vacuole, where they differentiate into small round-shaped amastigotes. Later, amastigotes exit the parasitophorous vacuole into the cell cytoplasm, where they proliferate by binary fission until the cell fills with these replicative forms [188]. At this point, some amastigotes elongate, regain a long flagellum, and differentiate into non-replicative trypomastigotes, which induce host cell lysis. Once released, trypomastigotes can infect other cells or enter the blood and disseminate to distant tissues. These trypomastigotes in the bloodstream can be taken up by triatomine insects during a blood meal. Inside the insect midgut, trypomastigotes become replicative epimastigotes. Finally, epimastigotes migrate to the insect hindgut, where they differentiate into metacyclic trypomastigotes [182,189]. More recently, other parasite stages have been suggested including quiescent forms of amastigotes [186].

### 5.1. Innate Immune Response against T. cruzi

Cells of the innate immunity, mainly phagocytes including macrophages, neutrophils, and dendritic cells, constitute the first line of defense against invading *T. cruzi* parasites [190]. In the acute phase, macrophages recognize and phagocytose parasites. Within the phagolysosome, parasites are destroyed by ROS and also reactive nitrogen species (RNS) [191,192]. However, *T. cruzi* has peroxidase and superoxide dismutase enzymes, which allow it to survive within the macrophage [166]. This allows the parasite to continue its invasion of other cells. In addition, *T. cruzi* antigens can be recognized by TLR2, and in response, macrophages secrete cytokines such as IL-1, IL-12, and TNF-α, which promote inflammation and induce activation of other immune cells such as T cells [180].

In addition to macrophages, neutrophils have always been observed in *Trypanosoma* infection sites [30]. Therefore, it is very surprising that there are very few reports exploring the role of neutrophils in Chagas disease. This may be in part due to the old idea that neutrophils were only a reflection of inflammation and also to the fact that the acute phase has nonspecific symptoms. Thus, many cases of *T. cruzi* infection go undetected. Despite this, it is becoming evident that neutrophils have an important role in Chagas disease.

### 5.2. Dual Role of Neutrophils in Chagas Disease

In the acute phase of Chagas disease, neutrophils were always associated only with inflammation. Nevertheless, old reports showed that the presence of neutrophils in cardiac lesions of Chagas disease patients correlated with the severity of the disease [193]. Also, neutrophils stimulated by *T. cruzi* amastigotes were capable of causing damage to cardiac cells [194]. However, neutrophils could kill *T. cruzi* trypomastigotes by antibody-dependent cell-mediated cytotoxicity [195] and also by phagocytosis and myeloperoxidase activity and ROS [196]. Thus, neutrophils seemed to be capable of eliminating *T. cruzi* parasites but also to exacerbate chronic disease.

More recently, in a mouse model of Chagas disease, specific depletion of neutrophils with anti-Ly6G^+^ antibody resulted in increased parasitemia and serum interferon (IFN)-γ concentration, leading to increased liver pathology [197]. This suggested a protective role for neutrophils. However, neutrophils seem to be able to increase or decrease the severity of the disease. Neutrophil depletion in BALB/c mice resulted in exacerbation of the disease with reduced expression of mRNA for Th1 cytokines. In contrast, depletion of neutrophils in C57BL/6 mice resulted in resistance to the disease with enhanced expression of Th1 cytokines [198]. In addition, in vitro co-cultures of BALB/c neutrophils with *T. cruzi*-infected peritoneal macrophages resulted in increased production of anti-inflammatory mediators such as prostaglandin-E2 (PGE2) and TGF-β, leading to increased replication of the parasites. In contrast, co-cultures of C57BL/6 neutrophils with infected macrophages resulted in the production of inflammatory mediators such as TNF-α and nitric oxide (NO), leading to decreased numbers of trypomastigotes [199].

These observations reveal that resistance or susceptibility to *T. cruzi* infection in these animal models is regulated in part by neutrophil functions. However, the mechanisms involved are still unknown. One possible way for controlling *T. cruzi* infection was revealed while using IL-17A^−/−^-deficient mice. These mice, when infected with *T. cruzi*, had lower survival rates than wild mice, due to increased parasitemia in several peripheral organs [200]. Mechanistically, IL-17 receptor (IL-17RA) is required for the recruitment of regulatory IL-10-producing neutrophils that destroy the parasite and control inflammatory responses [197]. In this way, IL-17 seems to be important for neutrophil activation required for killing *T. cruzi* parasites (Figure 4).

As suggested by the previous reports, neutrophils seem to have a protective role during the acute phase of Chagas disease. However, these cells also seem to be detrimental during the chronic phase of the disease, in which about 30% of patients develop cardiomyopathy, a condition generated by extracellular matrix remodeling. The MMP enzymes, together with some cytokines, are responsible for matrix remodeling. Neutrophils from patients with indeterminate Chagas disease produced more MMP-2, TGF-β, and IL-10 than neutrophils from patients with chronic cardiac disease. The latter cells produced more MMP-9, TNF-α, and IL-1β [201]. In addition, mice infected with *T. cruzi* and treated with apocynin, an inhibitor of NADPH oxidase, presented a reduction in myocarditis [202]. Together, these reports suggest that neutrophils may be involved in cardiac muscle remodeling and contribute to establishing the clinical forms of Chagas disease.

### 5.3. Neutrophil Functions against T. cruzi Parasites

#### 5.3.1. Phagocytosis

We have mentioned that neutrophils from healthy individuals are capable of recognizing *T. cruzi* amastigotes, phagocytose them, and destroy most of them within phagolysosomes [196] (Figure 4). Moreover, neutrophils from patients with the chronic cardiac form of Chagas disease displayed enhanced phagocytosis of *T. cruzi* parasites in vitro [203]. However, in another study, neutrophils from patients with indeterminate and cardiac forms of the disease presented comparable phagocytic capacity; although neutrophils from patients with indeterminate disease displayed a lower ability to produce cytokines, such as IL-17, IFN-**γ**, IL-4, and IL-10 [204]. The differences in both reports highlight the complex interaction of these parasites with neutrophils. Still, together they confirm that neutrophils are able to recognize, phagocytose, and eliminate *T. cruzi*.

#### 5.3.2. Reactive Oxygen Species (ROS)

Similar to macrophages, it is believed that neutrophils can kill *T. cruzi* parasites after phagocytosis via the production of ROS within the phagolysosome. This idea is reinforced by the fact that *T. cruzi* has very efficient antioxidant mechanisms to deal with the oxidative burst [166]. Still, there is no direct evidence that neutrophils can eliminate these parasites via ROS (Figure 4). Moreover, the nuclear factor, erythroid-derived 2, like 2 (NRF2), regulates antioxidant mechanisms. In mice infected with *T. cruzi*, NRF2 induction with cobalt protoporphyrin resulted in reduced parasitemia, and exogenous expression of NRF2 also reduced macrophage parasitism [205], suggesting that oxidative stress contributes to parasite persistence in host tissues. In addition, mice deficient in NADPH oxidase (phox^-/-^) and infected with *T. cruzi* had similar parasitemia and similar levels of IFN-γ and TNF-α in serum as those of wild-type control mice [206]. However, all phox^-/-^ mice died between days 15 and 21 after inoculation with the parasite, while 60% of wild-type mice survived 50 days after infection [206]. Hence, whilst ROS from phagocytes did not play a critical role in parasite control in the phox^-/-^ mice, its production still had a protective effect during infection with *T. cruzi*. Authors suggested that this effect was probably the control of blood pressure decline during infection [206]. In contrast, mice infected with *T. cruzi* and treated with apocynin, an inhibitor of NADPH oxidase, showed a reduction in myocarditis [202]. Together, these reports indicate that our understanding of the role of ROS for controlling *T. cruzi* infections is still very limited, and further research in this area is required.

#### 5.3.3. Neutrophil Extracellular Traps (NETs)

In the case of *T. cruzi*, it was found that both live and dead parasites could induce NET formation, after parasites were recognized by TLR-2 and TLR-4 [207] (Figure 4). These NETs efficiently trapped the parasites, but they did not kill them. However, NETs interfered with infectivity and pathogenicity [207]. Similarly, neutrophils from dogs and opossums responded to *T. cruzi* parasites by releasing NETs. These NETs ensnared, but again did not kill, the parasites [208]. Therefore, NETs do not seem to play a major role in elimination of the parasite from the blood, and consequently *T. cruzi* can survive in multiple niches of the body. Nevertheless, neutrophil activation and excessive NET formation can contribute to the destruction of surrounding tissues [209]. There is no information about the presence of NETs in the heart of patients with Chagas disease cardiomyopathy. Clearly, NETs can trap parasites, but their effect on the different forms of the disease are still controversial.

From these reports, it is evident that neutrophils can recognize *T. cruzi* parasites. However, it is still not clear to what extent neutrophils can destroy *T. cruzi* parasites and may also affect the pathogenicity of the parasite. Further research is necessary to elucidate the mechanisms that allow neutrophils to prevent or to allow progression of Chagas disease.

## 6. Amoebiasis

Amoebiasis is a disease caused by another protozoan parasite, the amoeba *Entamoeba histolytica* [210]. Amoebiasis is a serious public health problem in many parts of the world, particularly in tropical zones where this parasite is endemic [211]. According to The Global Burden of Diseases Study of 2016, it is estimated that over 123 million new cases of infection with *E. histolytica* are reported worldwide [212]. Amoebiasis is responsible for more than 26,000 deaths annually and 2.5 million disability-adjusted life year (DALY) cases [213]. *E. histolytica* infections remain a heavy burden among children younger than 5 years, particularly in areas with a low sociodemographic index [213]. However, a significant increasing trend for new infections is found among adults of all ages in regions with a high sociodemographic index [213]. In addition to the more pathogenic *E. histolytica*, there is also another amoeba, *Entamoeba dispar*, that can infect humans and other primates, causing, in rare instances, disease [214]. *E. histolytica* invades the intestinal tract, where it can live feeding on bacteria without causing any symptoms [215,216]. However, in some cases for reasons not completely elucidated, amoebas can penetrate the intestinal barrier triggering diarrhea, dysentery, and colitis. From the intestine, amoebas can invade other organs, principally the liver via the portal vein, where the parasite forms amoebic liver abscesses [217]. An interesting feature of amoebic liver abscesses is that they are 10 times more frequent in men than in women [218,219]. The reasons for this gender bias are not understood. One possible cause is complement activation. Serum from women was more effective in complement-mediated killing *E. histolytica* trophozoites than serum from men [220]. Another possible cause is testosterone. In mice, higher concentrations of testosterone increased the susceptibility to amoebic liver abscess by inhibiting the secretion of IFN-γ [221,222]. In rare cases, amoebas can also invade other organs such as lungs, heart, and even the brain. In these organs, amoebiasis is associated with a high mortality rate [223].

People get infected with *Entamoeba histolytica* through fecal–oral dissemination [224]. In contaminated water and food, there are *E. histolytica* dormant cysts which are resistant to the environment [225]. When ingested, the protective covering of cysts allows them to pass through the stomach unharmed. In the intestinal tract, excystation takes place, and the vegetative form of the parasite, the trophozoite, is liberated [226]. Trophozoites colonize the outer mucus layer of the large intestine. When the number of trophozoites increase, they group and form new cysts [227], which are released in feces [223].

In certain conditions, not completely known, more pathogenic trophozoites can destroy the mucosal layer, adhere to the epithelium, and invade tissues, causing disease [223,228,229]. The factors determining the pathological behavior of amoebas are not known. But, dysbiosis (changes in the microbiome) is probably a major factor. Other factors associated with disease severity may include, on the side of the host, malnutrition, pregnancy, cancer, alcoholism, corticosteroid use, lack of urban services [215,230], and even the host–gut microbiome [231]. On the side of the parasite, another factor for disease severity is the great genomic variability that exists among *E. histolytica* strains with different geographic mobility [232,233]. Under dysbiosis conditions, more pathogenic amoebas release glycosidases and proteinases that degrade the mucous layer [234]. Particularly, the cysteine proteinase EhCP-A5 may be the main proteinase in this process, since it has been found to be important for pathogenesis [235]. After degradation of the mucous layer, trophozoites adhere directly to epithelial cells using their galactose (Gal) and N-acetyl-D-galactosamine (GalNAc) lectin [225,228,236]. Then, trophozoites can break the epithelium by releasing cytotoxic molecules, such as amoebapores, cysteine proteinases, and phospholipase A2 [237]. In addition, trophozoites can kill epithelial cells by direct cell-contact mechanisms, which are not completely understood, that include apoptosis and trogocytosis [13]. Trophozoites induce apoptosis in part by increasing the intracellular calcium concentration and by triggering an efflux of potassium ions [238]. Also, trophozoites can cause cell death by trogocytosis [119]. Once trophozoites break the epithelium, they can move into the extra-intestinal space and disseminate to other organs. Epithelium cell death and degradation of the extracellular matrix contribute to initiate inflammation of the tissues [237]. In addition, the amoeba EhCP-A5 can bind to αvβ3 integrins on goblet cells, activating the NLRP3 inflammasome [239]. This inflammatory response leads to production of IL-8 [240], which is important for recruiting neutrophils to control parasite invasion.

### 6.1. Role of Neutrophils in Amoebiasis

When *E. histolytica* trophozoites invade the extra-intestinal space, strong inflammation develops, and many neutrophils infiltrate the affected tissue [241] (Figure 5). If trophozoites disseminate to other organs, such as the liver, again a strong inflammatory reaction is observed with many infiltrating neutrophils [242]. Early reports described neutrophils moving vigorously around amoebas [243], and neutrophils could directly kill these parasites in vitro [244]. More recently, it was reported that lower numbers of neutrophils result in more severe amoebiasis [245,246,247,248] and that the presence of neutrophils also reduces amebic colitis [249]. Consequently, it is largely recognized that neutrophils play a protective role against amoebas [245]. However, there are reports indicating that amoebas induce neutrophil death, leading to the release of their lytic enzymes and tissue damage. This, in turn, facilitates amoeba invasion and development of amoebiasis [250,251,252]. Hence, the role of neutrophils in amoebiasis remains controversial.

During inflammation, neutrophils exit the blood and migrate to the affected tissue, where they destroy small microorganisms through phagocytosis [19]. For large microorganisms, such as amoebas, phagocytosis is not possible. Thus, neutrophils cooperate by swarming around the large microorganism [253,254]. In the case of amoebiasis, neutrophil swarming has not been reported. However, in vitro neutrophils have been observed to actively cover *E. histolytica* trophozoites [255]. This suggests that neutrophils indeed implement swarming around this parasite. Neutrophil swarming is not easily revealed in vitro. But, novel techniques, such as microfluidics [111,256] and microscale swarming arrays [257], will contribute to reveal whether neutrophils indeed control large amoeba parasites via swarming.

### 6.2. Neutrophil Functions against E. histolytica

The neutrophil mechanisms for controlling an amoebic infection are only partially understood. It is believed that amoebas are killed by ROS but not very efficiently, and in consequence, amoebas can then destroy the neutrophils, leading to more tissue damage. This view is rapidly changing as discussed next.

#### 6.2.1. Reactive Oxygen Species (ROS)

As mentioned, the current belief is that ROS produced by neutrophils can control amoebas [258,259]. This idea is based on in vitro experiments showing that H_2_O_2_ could induce an apoptosis-like death of trophozoites [260], and on the fact that amoebic peroxiredoxin could degrade ROS generated by leukocytes [259,261]. However, there are no reports indicating that ROS generated by neutrophils can directly destroy amoebas. In fact, recent in vivo data from mouse models of amoebic liver abscess show that ROS exacerbate liver tissue damage without reducing parasitemia. When mice were treated with antioxidants (e.g., ascorbic acid), they had a significant decrease in liver lesions [262,263]. Additionally, recent reports showed that human neutrophils did not produce ROS when in contact with pathogenic *E. histolytica* [264,265], or with non-pathogenic *E. dispar* [255]. Therefore, contrary to the traditional view, it is probable that neutrophils use other antimicrobial functions for controlling amoebiasis.

#### 6.2.2. Neutrophil Extracellular Traps (NETs)

Phagocytosis of amoebas is impossible due to the large size of these protozoan parasites. However, NETosis is another strategy of neutrophils to control the spreading of microorganisms. In the case of amoebas, neutrophils can indeed form NETs in response to *E. histolytica* trophozoites [264,266] (Figure 5). This more pathogenic amoeba triggers NETosis through a signaling pathway involving Raf/MEK/ERK, but not PKC or ROS [264,265]. Neutrophils in direct contact with trophozoites released NETs in an explosive manner around the amoebas [255]. Several neutrophils released NETs until the trophozoite was completely covered with NETs and was immobilized [255]. Moreover, *E. histolytica* trophozoites inhibited the neutrophil oxidative burst in a dose-dependent manner, and inhibition of mitochondrial ROS (by the mitochondria-specific ROS scavenger mitoTEMPO) also did not affect NET formation [267]. These data reinforced the idea that *E. histolytica*-induced NETosis was independent of ROS. Surprisingly, however, in the presence of ROS-deficient amoebas (obtained by pre-treatment with pyrocatechol) neutrophils showed a significant reduction in NET formation [267]. This created a conundrum of how do amoebas inhibit neutrophil oxidative burst and, at the same time, provide ROS for NET formation? The complete answer is not known, but few clues are emerging. Amoebas released extracellular vesicles (EVs), which were combined into neutrophils, delivering their cargo into the cell. This resulted in considerable inhibition of the oxidative burst and NET formation from neutrophils stimulated by PMA, ionophore A23187, or the amoeba itself [268]. Amoebic EVs contained ROS and were able to transfer them to neutrophils, suggesting that this may be a way to provide ROS for NET formation. However, amoebic EVs still had a suppressive effect on NETosis induced by other stimuli [268]. Further research is needed to elucidate the mechanisms by which amoebic EVs prevent respiratory burst and NETosis. Although extracellular vesicles are an important mode of communication between parasites and immune cells, this topic is beyond the scope of the present review. We direct readers to an excellent recent review on the topic [269].

Another important observation was that the less-pathogenic *Entamoeba dispar* did not trigger NET formation [255]. This finding suggests that neutrophils can distinguish between pathogenic *E. histolytica* and less-pathogenic *E. dispar* and activate NETosis only in response to pathogenic amoebas. How neutrophils can differentiate between pathogenic and non-pathogenic amoebas is not known. However, in the presence of Gal or GalNAc, NET formation was inhibited [255]. This suggested that the mechanism must likely involve a receptor that recognizes sugar moieties with an axial HO- group at carbon 4 of a hexose. Further research is needed to confirm this hypothesis.

NETs released around pathogenic *E. histolytica* not only prevented amoebas from moving, but were also able to kill the trophozoites [255]. The mechanism for amoeba killing is not yet clear, but it may require some of the neutrophil granule proteins that decorate NETs. Myeloperoxidase (MPO) may participate in killing amoebas. Using hamsters (susceptible) and Balb/c mice (resistant) models of amoebic liver abscess, it was found that inhibition of MPO resulted in mice with larger abscesses [270]. Also, in vitro, mouse neutrophils produced more NETs and MPO than hamster neutrophils did [271], and inhibition of MPO resulted in larger amounts of viable amoeba [272]. Histone proteins on NETs have also antimicrobial properties. As mentioned in the previous section, histones on NETs were able to affect the viability of *L. amazonensis* promastigotes [169]. Thus, it is likely that histones may also have microbicidal activity against amoebas. Since NETs can kill *E. histolytica* trophozoites, the participation of histones in amoeba killing should be examined in future experiments.

The classical view of amoebas promoting neutrophil death by apoptosis [273] is in conflict with the recent reports described above. In these recent studies, apoptosis of neutrophils in contact with *E. histolytica* trophozoites was not detected. Neutrophils did not expose phosphatidylserine on their membrane [264,265]. Therefore, a new paradigm for *E. histolytica* infection is developing. Invading trophozoites are surrounded by neutrophils, and only neutrophils in direct contact with pathogenic trophozoites activate NETosis. Then, NETs are released around the amoebas to prevent their movement and dissemination. Thus, dead neutrophils around amoebas appear not to be killed by trophozoites but instead are in fact neutrophils undergoing NETosis.

## 7. Conclusions

Why, in some cases, neutrophils can control the parasite and in others the infection proceeds resulting in disease, is a central question that needs future research. Yet, there are many other particular questions that need to be addressed in order to elucidate the complex interaction of neutrophils with protozoan parasites. For example, in the case of *Plasmodium*, it is still uncertain to what extent phagocytosis of free parasites helps in controlling parasite proliferation. Also, because multiple mechanisms, both from the parasite and also from the mosquito, are involved in inhibition of neutrophil antimicrobial functions in malaria, it is important to investigate each mechanism separately in order to obtain a complete picture of how *Plasmodium* avoids neutrophils and causes disease. In the case of *T. vaginalis*, although neutrophils can kill parasites by trogocytosis, many parasites survive and perpetrate the infection. Finding a way to activate neutrophils so that they could eliminate more parasites, without increasing the severity of the disease, could have a major impact in the treatment of trichomoniasis. In the case of Leishmania, little is known about how the parasite inhibits phagolysosome formation in neutrophils after phagocytosis of promastigotes. In particular, how differences in phagolysosome maturation affect the promastigote to amastigote transition for each *Leishmania* species is also an important question. Also, much is unknown about how the different species of *Leishmania* resist the various antimicrobial mechanisms of neutrophils. In addition, the biology of *Leishmania* parasitophorous vacuoles in neutrophils has not been properly studied. Another significant unsolved issue about NETosis is how neutrophils choose between carrying phagocytosis out or releasing NETs. Our current understanding is that phagocytosis and NETosis are mutually exclusive [173]. In the case of *Leishmania*, NETs can trap and kill the parasites. However, phagocytosis is more useful to the parasite in order to continue the infection. Thus, it would be very important to decipher how *Leishmania* parasites induce neutrophils to perform phagocytosis over NETosis. In the case of Chagas disease, *T. cruzi* parasites clearly activate multiple neutrophil functions including phagocytosis, ROS generation, and NET formation [166,196,207]. However, in chronic cardiac disease, activated neutrophils seem to be more detrimental than helpful. Future research will have to look into the different ways in which neutrophils recognize the parasite and become activated. In the case of amoebas, neutrophils can release NETs around *Entamoeba histolytica* trophozoites [264,266], until parasites are completely covered with NETs and are immobilized [255]. Also, because NETs could kill *E. histolytica* trophozoites [255], the participation of histones in amoeba killing should be examined in future experiments. Interestingly, while the pathogenic *E. histolytica* can induce NETosis [264,266], the less-pathogenic *E. dispar* did not trigger NET formation [255]. How neutrophils can differentiate between pathogenic and less-pathogenic amoebas is not known.

## Figures and Tables

**Figure 1 microorganisms-12-00827-f001:**
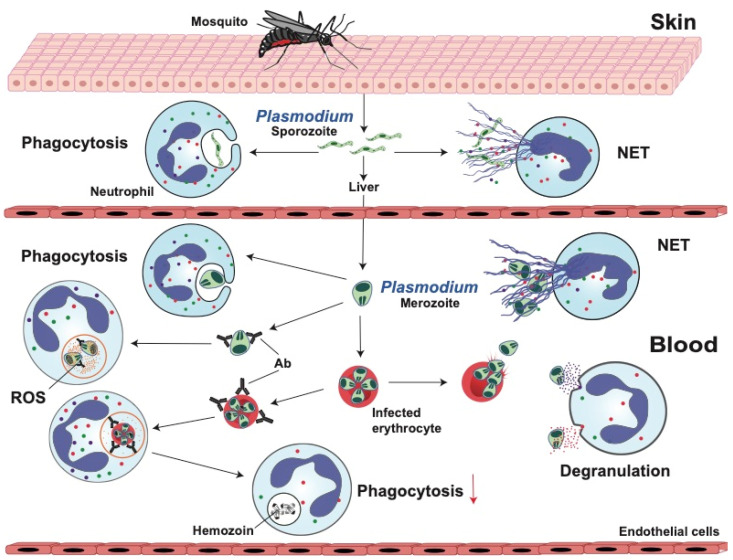
**Interactions of neutrophils with *Plasmodium* parasites**. During a blood meal, a female *Anopheles* mosquito introduces *Plasmodium falciparum* sporozoites into the human host. Neutrophils are the first leukocytes to respond to the *Plasmodium* infection site, where they can phagocytose free sporozoites. Also, neutrophils release neutrophil extracellular traps (NETs) that ensnare free sporozoites. Sporozoites mature into merozoites as they travel to the liver and ultimately to the bloodstream. *Plasmodium* merozoites infect erythrocytes, where they continue dividing until the erythrocyte is destroyed, releasing more merozoites. The continuous destruction of erythrocytes is the main cause of malaria. Neutrophils are the first leukocytes to respond to the *Plasmodium* infection site, where they can phagocytose free sporozoites. Also, neutrophils release NETs that ensnare free sporozoites. In the blood, neutrophil phagocytosis of free or antibody (Ab)-coated merozoites can take place. Inside the neutrophil, parasites can be killed by reactive oxygen species (ROS). Neutrophil phagocytosis of infected erythrocytes is observed more frequently, particularly if erythrocytes are opsonized with antibodies (Ab). Neutrophils that have ingested infected erythrocytes accumulate hemozoin (malaria pigment), which is the end product of hemoglobin digestion by parasites. Hemozoin is able to inhibit further phagocytosis (down red arrow). In addition, neutrophils can control parasite burden by degranulation. Granule proteins such as MMP-9 (matrix metallopeptidase 9) can damage merozoites directly. Finally, *Plasmodium*-infected erythrocytes can induce NET formation by releasing MIF (macrophage migration inhibitory factor), uric acid crystals, or heme (a hemoglobin breakdown product). Thus, NETs are an important mechanism to control parasite dissemination.

**Figure 2 microorganisms-12-00827-f002:**
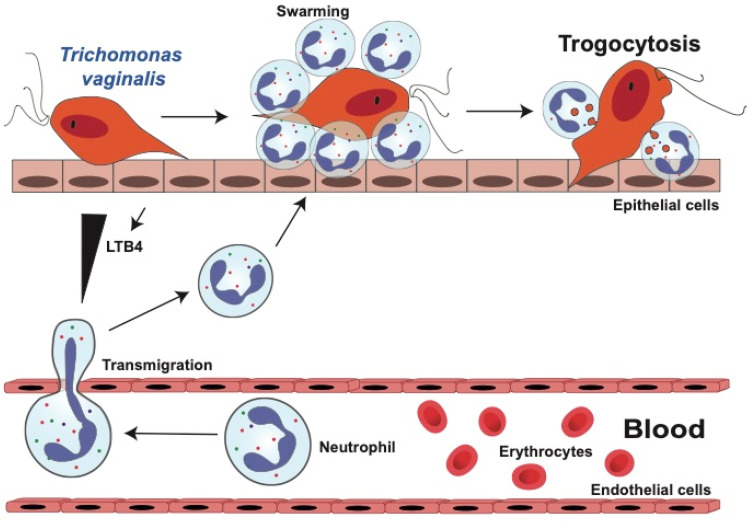
**Interactions of neutrophils with *Trichomonas vaginalis*.** *T. vaginalis*, an extracellular parasite, adheres to epithelial cells in the urogenital tract. This initial interaction induces epithelial cells at the infection site to produce LTB4 (leukotriene-B4), which promotes extravasation of neutrophils. Because *T. vaginalis* also produces LTB4, neutrophils follow this cue towards the parasites. In response to LTB4, neutrophils display a swarming behavior around the parasites. Neutrophils then kill *T. vaginalis* by taking small pieces, i.e., “bites”, of the parasite membrane, a process known as trogocytosis.

**Figure 3 microorganisms-12-00827-f003:**
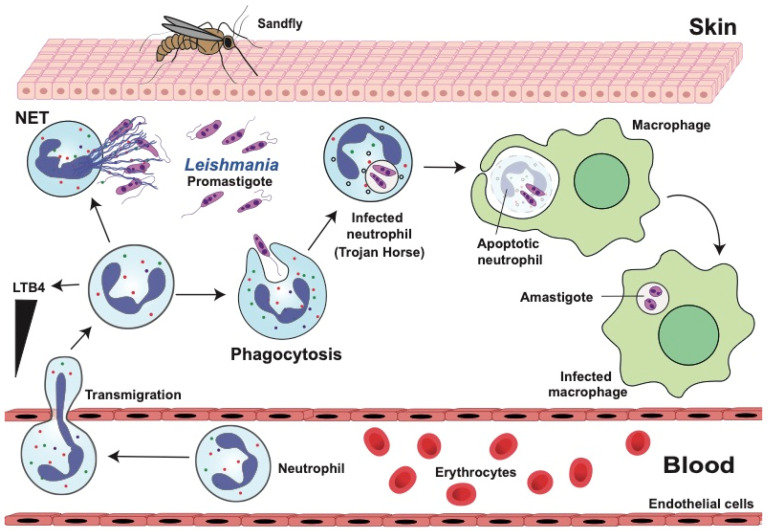
**Interaction of neutrophils with *Leishmania* parasites.** *Leishmania* protozoa are obligated intracellular parasites. *Leishmania* promastigotes are transmitted to humans when a female sandfly of *Phlebotomus* (in Africa and Asia) or *Lutzomia* (in New World) genera takes a bloodmeal. Promastigotes are deposited in the skin epidermis or the upper layer of the dermis, where they are rapidly internalized by phagocytic cells, mainly neutrophils. This initial interaction induces neutrophil activation, resulting in production of LTB4 (leukotriene-B4), which promotes extravasation of more neutrophils. Neutrophils can also produce neutrophil extracellular traps (NETs) to prevent parasite spreading. Within the neutrophil, promastigotes transform into amastigotes, which rapidly replicate and disseminate to other cells. Although *Leishmania* promastigotes can be killed by phagocytosis, destruction of parasites varies among *Leishmania* species. For example, *L. braziliensis* and *L. donovani* are susceptible, while *L. amazonensis* and *L. mexicana* can survive within neutrophils; most likely by preventing phagolysosome maturation. In addition, parasites stimulate the apoptosis of neutrophils. Then, infected apoptotic neutrophils are taken up by macrophages, the parasite final host cells. This process, which allows parasites to infect macrophages unnoticed, has been described as neutrophils being a “Trojan horse” into macrophages.

**Figure 4 microorganisms-12-00827-f004:**
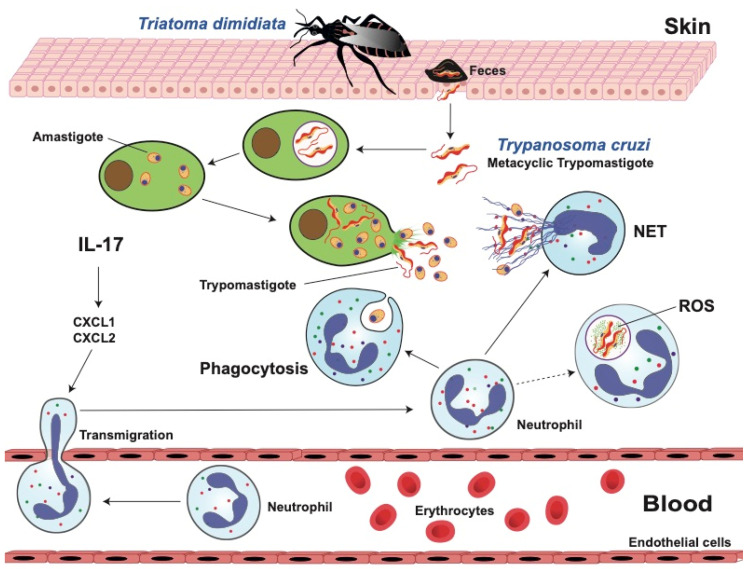
**Interaction of neutrophils with** ***Trypanosoma cruzi*** **parasites**. *T. cruzi* metacyclic trypomastigotes present in feces from a *Triatoma dimidiata* insect penetrate the skin through the insect bite wound. Metacyclic trypomastigotes infect nucleated cells entering into a parasitophorous vacuole, where they differentiate into amastigotes. Amastigotes exit the vacuole into the cell cytoplasm where they divide. Some amastigotes also differentiate into non-replicative trypomastigotes, which induce host cell lysis. IL-17 recruits and activates neutrophils from the blood stream via CXCL1 and CXCL2 chemokines. Neutrophils phagocytose and kill amastigotes. Neutrophils also secrete neutrophil extracellular traps (NETs) to trap and kill trypomastigotes and amastigotes. There is no direct evidence that neutrophils can eliminate *T. cruzi* parasites via reactive oxygen species (ROS).

**Figure 5 microorganisms-12-00827-f005:**
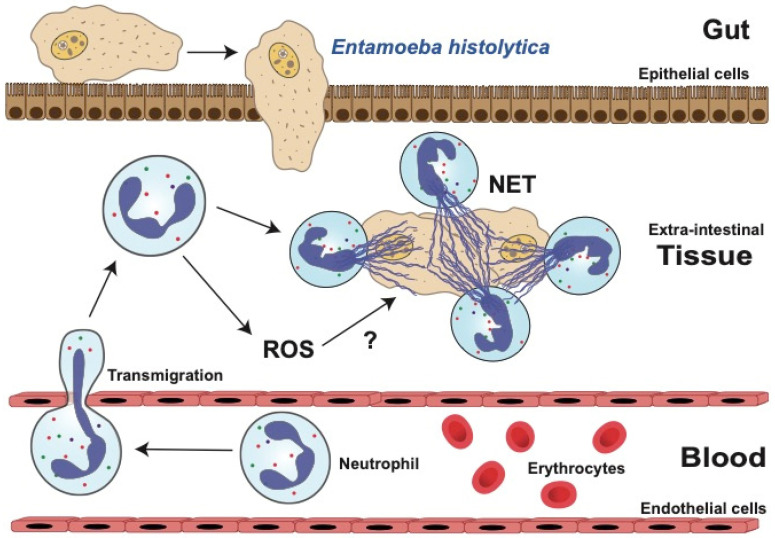
**Interaction of neutrophils with pathogenic *Entamoeba histolytica***. *E. histolytica* trophozoites colonize the outer mucus layer of the large intestine (gut), where they can live feeding on commensal bacteria without causing any symptoms. However, in some cases for reasons not completely elucidated, amoebas can penetrate the intestinal barrier, triggering disease. Once trophozoites break the epithelium, they can move into the extra-intestinal space, inducing a strong inflammation that recruits many neutrophils into the affected tissue. Neutrophils can then actively cover *E. histolytica* trophozoites and form neutrophil extracellular traps (NETs). Several neutrophils release NETs, in an explosive manner, around a single amoeba until the trophozoite is immobilized and probably also killed.

## Data Availability

No new data were created or analyzed in this review.

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
