# Peer review of "Neutrophils versus Protozoan Parasites: Plasmodium, Trichomonas, Leishmania, Trypanosoma, and Entameoba"

_microorganisms, 2024, doi:10.3390/microorganisms12040827_

Round 1

Reviewer 1 Report

Comments and Suggestions for Authors

In this review, the authors summarized several protozoan diseases and described the novel view of how neutrophils are involved in protection from these parasites. Also, recent evidence that neutrophils play a double role in these infections participating in the control of the parasite and the pathogenesis of the disease was presented. This paper was well-prepared and displayed interesting results on the role of neutrophils in several protozoan diseases. Therefore, I suggest the acceptance of this paper after minor revision as follows:

1. Due to the importance of Chagas disease (Trypanosoma cruzi) and sleep sickness (Trypanosoma brucei), I suggest including a topic regarding the effects on Trypanosoma sp.

2.   Please, change Leishmania chagasi for Leishmania infantum chagasi

3. Please, change the sentence “Lutzomia (in America)” to “Lutzomia (in New World)”

Author Response

Reviewer 1

In this review, the authors summarized several protozoan diseases and described the novel view of how neutrophils are involved in protection from these parasites. Also, recent evidence that neutrophils play a double role in these infections participating in the control of the parasite and the pathogenesis of the disease was presented. This paper was well-prepared and displayed interesting results on the role of neutrophils in several protozoan diseases. Therefore, I suggest the acceptance of this paper after minor revision as follows:

  1. Due to the importance of Chagas disease (Trypanosoma cruzi) and sleep sickness (Trypanosoma brucei), I suggest including a topic regarding the effects on Trypanosoma sp.
  2.   Please, change Leishmania chagasi for Leishmania infantum chagasi
  3. Please, change the sentence “Lutzomia (in America)” to “Lutzomia (in New World)”

Thank you very much for your comments and suggestions. We did not include a section on Trypanosoma because there is very little information about neutrophils and these parasites. However, we see the relevance of this topic. Therefore, we have reviewed the manuscript by adding a new section describing Trypanosama cruzi (New section 5)

The terms suggested have been changed as suggested.

Reviewer 2 Report

Comments and Suggestions for Authors

The topic in general is interesting and several of the chapters are nicely written.

However:

The overview of neutrophils is definitely too superficial, important functional and immunological properties of neutrophils should be described in detail in the introduction, including all neutrophil functions later mentioned in the review and all other functions. The literature cited is not state of the art.

The consequence of this is a high degree of redundancy in the description in the individual chapters e.g. Extravasation, NET formation. Redundancies are also in the Figures.

The headlines of the subchapters are structured very differently. While they are grouped according to neutrophil function (e.g. ROS production, phagocytosis) in the case of Plasmodium neutrophil interaction, chapters are structured very differently for other parasites, e.g. Trichomonas/ Leishmania. Better consistency is advisable here.

Microvesicles or extracellular vesicles (which is the recommended term, please refer to the MISEV guidelines published in 2014, 2018 and 2023) are one of the main modes of communication between parasites and immune cells. This was discussed in the chapter on one parasite (Trichomonas), but not for others (Leishmania, Entamoeba), even though there is current literature on this.

Therefore, a generally similar or more uniform structure of the content of the chapters would be helpful.

The rationale for choosing to elaborate on bacterial co-infections in the context of Plasmodium infection but not other parasites is not apparent in this review.

Discussion of current literature on in vitro and in vivo data on neutrophils and E. histolytica is lacking.

Minor:

english needs revision, particularly the use of prepositions in some sentences

table of content would be helpful

Lane 72: symptoms resulting from Malaria infection are not properly described (e.g. for severe Malaria, cerebral malaria, coma...)

Lane 178: It is not apparent to the reader how NET formation and parasite sequestration are linked. Please elaborate further.

Lane 181: Interaction of NET formation and degranulation should be explained better here.

Chapter 3, epidemiological data are missing, compared to other sections

Lane 272 – 274: Redundant information also described in the above paragraph (chapter 3)

Lane 299 – 300: Inconsistent tense

Lane 304 – 307: Information on trogocytosis of sperm not relevant in the context of this review (or relevance needs to be elaborated).

Fig. 2, for which reason erythrocytes are shown?

Lane 391: Which chemokines are released for the recruitment? (chemokines involved in the recruitment of neutrophils should also be mentioned in the introduction); Which factors in the sandfly saliva play a role?

Lane 293: Which gel compartments are produced by the promastigote parasite? Would you comment on this?

Lane 397-398: Sentence unnecessary

"4.1. Neutrophils in Leishmania, this is the general topic of the review, why is this used as headline here? Again, standardization of the headings in the chapters would be appropriate "

Lane 410 which cytokines?

LTB4 this is shown for Trichomonas (Fig.2), is this relevant for Leishmania as well? LTB4 is produced by epithelial cells, also by neutrophils?  Please describe more specifically, Reference is missing.

Lane 422: here it would be helpful to include knowledge about the uptake of apoptotic neutrophils by macrophages, are there differences between Leishmania -infected apoptotic neutrophils or general uptake of apoptotic cells ?

This section (427-434) should be revised and described more precisely. Promastigotes of L. amazonensis are killed, and only amastigotes survive. How is this possible? This is not clear.

Lane 434. Promastigote and amastigotes wording should be used here instead of parasite, this mixes the two forms and is confusing for the reader in this part.

Lane 445-446: Sentence unnecessary

Lane 522 – 529: Conclusion statement of all subchapters on Leishmania not fitting under 4.3 here.  Revise chapter structure

Lane 533 – 535: For the most recent data on estimated amebiasis-related deaths, please consult the Global Burden of Disease Report 2016 (https://www.sciencedirect.com/science/article/pii/S1473309918303621?via%3Dihub) instead of citing another review (Also, this data is not even stated in the review you cite!).

Lane 535 The simple differentiation between E. histolytica and E. dispar as pathogenic and nonpathogenic can no longer be maintained in this strict form in the beginning, please fuse with the "limitation" statement some lanes below (lane 556 and following). Moreover, problems arising from the strains for diagnosis are not relevant for this review article.

Lane 540: What do you mean by susceptible people? Currently, it is not known why Entamoeba histolytica becomes invasive and what the role of host susceptibility vs. parasite virulence is. Please phrase more carefully. (also applies to lane 601)

Lane 543: In the context of amoebic liver abscess formation, mention the male predominance 

Lane 549 – 550: Limit the use of ‘Then’ at the beginning of sentences

lane 539 and 551: redundant

lane 549 and 554: redundant

Lane 549: Intestinal track  Intestinal tract

Lane 556: Which other factors in addition to dysbiosis could be involved?

Lane 573 and following: You previously mentioned the role of IL-8 in the recruitment of intracellular neutrophils in the context of another parasite, it would be good to include this here as well.

Figure 4: Neutrophil functions other than NETs have previously been described in the context of Entamoeba-neutrophil interaction, please include them in the Figure (e.g. ROS).

Lane 628: this statement needs revision...the word "choice" is over-interpreted and, moreover, not properly assured

Lane:638-639: this statement should be revised, e.g.: Is there evidence that these sugar moieties are present in different amounts on pathogenic and non-pathogenic amoebae?

The recent work of César Díaz-Godínez, should be included in terms of "Microvesicles", E. histolytica and neutrophils" (Front Cell Infect Microbiol, 2022 10.3389/fcimb.2022.1018314)

Conclusion: You quickly summarize the main points for Plasmodium, Leishmania and Entamoeba but not for Trichomonas.

Lane 709 – 714: Unfitting final conclusion

References missing , please control the text carefully

lane 51

lane 192

lane 226

lane 316, or is this still refence 90?

lane 388

lane 404

lane 427

lane 540

lane 541

lane 549

lane 632

Comments on the Quality of English Language

As mentioned above, the quality of the english should be improved.

Author Response

Reviewer 2

We thank this reviewer for their careful reading of our manuscript and for the many suggestions to improve our report. We have revised our manuscript accordingly and include all of them in the new version. We think, our review is a more robust report now.

We next describe the revisions included.

"The overview of neutrophils is definitely too superficial, important functional and immunological properties of neutrophils should be described in detail in the introduction, including all neutrophil functions later mentioned in the review and all other functions. The literature cited is not state of the art."

" The consequence of this is a high degree of redundancy in the description in the individual chapters e.g. Extravasation, NET formation. Redundancies are also in the Figures."

We have expanded the introduction to include a more detailed description of antimicrobial neutrophil functions. At the same time, redundant text in the following sections was eliminated. Also, we revised and updated the reference list.

"The headlines of the subchapters are structured very differently. While they are grouped according to neutrophil function (e.g. ROS production, phagocytosis) in the case of Plasmodium neutrophil interaction, chapters are structured very differently for other parasites, e.g. Trichomonas/ Leishmania. Better consistency is advisable here."

We thank the reviewer for this observation. We have revised the manuscript in order to have better consistency describing first the neutrophil effects on the parasite and then the evasive mechanisms of the parasite.

"Microvesicles or extracellular vesicles (which is the recommended term, please refer to the MISEV guidelines published in 2014, 2018 and 2023) are one of the main modes of communication between parasites and immune cells. This was discussed in the chapter on one parasite (Trichomonas), but not for others (Leishmania, Entamoeba), even though there is current literature on this."

"Therefore, a generally similar or more uniform structure of the content of the chapters would be helpful."

We thank the reviewer for this observation. Although, extracellular vesicles are an important mode of communication between parasites and immune cells, this topic is beyond the scope of the present review. Therefore, further discussion on this topic has been eliminated from our manuscript. However, the reader is directed to an excellent review on the topic recently published.

"The rationale for choosing to elaborate on bacterial co-infections in the context of Plasmodium infection but not other parasites is not apparent in this review".

Bacterial co-infections are a major problem clearly associated with malaria. The same association has not been documented for other parasitic diseases. This is a problem related to neutrophil function. This idea has been added in the revised manuscript.

"Discussion of current literature on in vitro and in vivo data on neutrophils and E. histolytica is lacking."

Thank you for bringing this to our attention. More recent publications related to in vivo data on neutrophils and E. histolytica, have been added in the revised manuscript.

Minor:

"english needs revision, particularly the use of prepositions in some sentences"

We have revised the whole manuscript for proper use of the English language.

"table of content would be helpful"

We thank the reviewer for this suggestion. However, the journal format does not include a table of content. Thus, we did not include one.

"Lane 72: symptoms resulting from Malaria infection are not properly described (e.g. for severe Malaria, cerebral malaria, coma...) "

We have included a better description of malaria symptoms for the three forms of the disease

"Lane 178: It is not apparent to the reader how NET formation and parasite sequestration are linked. Please elaborate further."

Thank you for pointing out this issue. The text has been revised to make the connection between NET and parasite sequestration and malaria severity clearer.

"Lane 181: Interaction of NET formation and degranulation should be explained better here"

This sentence has been deleted.

"Chapter 3, epidemiological data are missing, compared to other sections"

Thank you for pointing this omission out. We have added a new paragraph including this information

"Lane 272 – 274: Redundant information also described in the above paragraph (chapter 3)"

Thank you for pointing this out . We have deleted the corresponding text.

"Lane 299 – 300: Inconsistent tense"

Thank you again for your careful reading. We have corrected the verb tense.

"Lane 304 – 307: Information on trogocytosis of sperm not relevant in the context of this review (or relevance needs to be elaborated)."

We agree with the reviewer. Text has been deleted.

"Fig. 2, for which reason erythrocytes are shown?"

Erythrocytes are the most abundant cells in blood. Few of them are shown in the figure just for illustration purposes. They convey the idea of whole blood, without distracting from the neutrophils.

"Lane 391: Which chemokines are released for the recruitment? (chemokines involved in the recruitment of neutrophils should also be mentioned in the introduction); Which factors in the sandfly saliva play a role?"

The chemokines involved are CXCL1, CXCL2, and CXCL5. They have been added in the revised manuscript. Also, the components in the sandfly saliva that may have a role have been explained in the revised manuscript. In addition, information of neutrophil-attracting chemoattractans has also included in the introduction of the revised manuscript.

"Lane 293: Which gel compartments are produced by the promastigote parasite? Would you comment on this?"

Thank you for bringing to our attention that this part was not clear. In the sand fly midgut secreted proteophosphoglycans from Leishmania form a biological plug known as the promastigote secretory gel (PSG), which blocks the gut and facilitates the regurgitation of infective parasites. We have included this information in the revised manuscript.

"Lane 397-398: Sentence unnecessary"

Thank you for the suggestion. The sentence has been deleted.

"4.1. Neutrophils in Leishmania, this is the general topic of the review, why is this used as headline here? Again, standardization of the headings in the chapters would be appropriate "

Thank you again for this suggestion. The headlines of the subchapters have been changed to have more consistency.

"Lane 410 which cytokines?"

The relevant cytokines are particularly IL-1β, TNF-α, TGF-β, and IL-6. These have been included in the revised manuscript.

"LTB4 this is shown for Trichomonas (Fig.2), is this relevant for Leishmania as well? LTB4 is produced by epithelial cells, also by neutrophils?  Please describe more specifically, Reference is missing."

Indeed, LTB4 is also relevant for neutrophil recruitment during Leishmania infections. This was clearly stated in our original manuscript, but it was not shown in figure 3. The figure has been revised to show LTB4. One relevant reference was included in the original manuscript, and a second more recent one has been added in the revised manuscript.

"Lane 422: here it would be helpful to include knowledge about the uptake of apoptotic neutrophils by macrophages, are there differences between Leishmania -infected apoptotic neutrophils or general uptake of apoptotic cells? "

This line corresponds to a figure legend, where things are described briefly. A more extensive description of macrophages ingesting apoptotic neutrophils (efferocytosis) is presented in the main text. Nothing has been reported about differences in uptake by macrophages of infected apoptotic neutrophils and just apoptotic neutrophils. Since, parasites induce neutrophils to enter in apoptosis, it seems that apoptotic signals, and not the parasite, are the trigger for efferocytosis.

"This section (427-434) should be revised and described more precisely. Promastigotes of L. amazonensis are killed, and only amastigotes survive. How is this possible? This is not clear."

Thank you for pointing this out. Promastigotes are killed more efficiently than amastigotes in neutrophils. The exact mechanism is not known, but it may be related to the different cytokines involved. This has been added in the revised manuscript.

"Lane 434. Promastigote and amastigotes wording should be used here instead of parasite, this mixes the two forms and is confusing for the reader in this part."

Thank you for the suggestion. The word parasite has been substituted by promastigotes and amastigotes in the revised manuscript.

"Lane 445-446: Sentence unnecessary"

Thank you for the suggestion. The sentence has been deleted.

"Lane 522 – 529: Conclusion statement of all subchapters on Leishmania not fitting under 4.3 here. à Revise chapter structure"

We agree with the reviewer. This paragraph has been deleted and the information included in the final conclusions section.

"Lane 533 – 535: For the most recent data on estimated amebiasis-related deaths, please consult the Global Burden of Disease Report 2016 (https://www.sciencedirect.com/science/article/pii/S1473309918303621?via%3Dihub) instead of citing another review (Also, this data is not even stated in the review you cite!). "

Thank you for the suggestion to include the most recent data on amebiasis-related deaths. The text has been corrected accordingly including new references

"Lane 535 The simple differentiation between E. histolytica and E. dispar as pathogenic and nonpathogenic can no longer be maintained in this strict form in the beginning, please fuse with the "limitation" statement some lanes below (lane 556 and following). Moreover, problems arising from the strains for diagnosis are not relevant for this review article."

We completely agree with the reviewer. We have corrected our manuscript to show that E. histolytica is more aggressive (pathogenic) than E. dispar, but that both amoebas can cause disease. This idea has been included in the whole section on amoebiasis in the revised manuscript.

"Lane 540: What do you mean by susceptible people? Currently, it is not known why Entamoeba histolytica becomes invasive and what the role of host susceptibility vs. parasite virulence is. Please phrase more carefully. (also applies to lane 601)"

The reviewer is right and we agree. The reasons Entamoeba histolytica to become invasive are not clear. We corrected our manuscript by deleting the phrase " susceptible people"

"Lane 543: In the context of amoebic liver abscess formation, mention the male predominance" 

Thank you for this suggestion. We have revised our manuscript to include this information.

"Lane 549 – 550: Limit the use of ‘Then’ at the beginning of sentences"

Thank for the suggestion. Text has been corrected accordingly.

"lane 539 and 551: redundant"

We agree. First sentence was edited. The second one was deleted.

"lane 549 and 554: redundant"

Thank you for the suggestions. The First sentence was edited. The second one was deleted.

"Lane 549: Intestinal track à Intestinal tract"

Thank you for your careful reading. The text was corrected.

"Lane 556: Which other factors in addition to dysbiosis could be involved?"

Thank you for the suggestion to include other factor related to pathogenicity. Other potential factors have been included in the revised manuscript.

"Lane 573 and following: You previously mentioned the role of IL-8 in the recruitment of intracellular neutrophils in the context of another parasite, it would be good to include this here as well."

We thank the reviewer for this suggestion. IL-8 has been included here.

"Figure 4: Neutrophil functions other than NETs have previously been described in the context of Entamoeba-neutrophil interaction, please include them in the Figure (e.g. ROS).

The reviewer is right than in the classical view, amoebas could be killed by neutrophils through ROS. However, this idea is not supported by new reports showing that the main neutrophil function against amoebas seems to be the release of NET. Nonetheless, we see the importance of it, and ROS have been included in the new revised figure 5.

"Lane 628: this statement needs revision...the word "choice" is over-interpreted and, moreover, not properly assured"

Thank you for pointing this out. We agree the idea is not clear. The statement has been deleted in the revised manuscript.

"Lane:638-639: this statement should be revised, e.g.: Is there evidence that these sugar moieties are present in different amounts on pathogenic and non-pathogenic amoebae?"

Thank you for pointing this issue. No, there is no evidence that these sugar moieties are present in different amounts on both amoebas. The data show that some free carbohydrates inhibit NET formation. This should help in the future to try identifying a putative receptor. Therefore, the text has been revised to indicate that this is just a working hypothesis.

"The recent work of César Díaz-Godínez, should be included in terms of "Microvesicles", E. histolytica and neutrophils" (Front Cell Infect Microbiol, 2022 10.3389/fcimb.2022.1018314) "

Thank you for the suggestion. This work has been included in the revised manuscript.

"Conclusion: You quickly summarize the main points for Plasmodium, Leishmania and Entamoeba but not for Trichomonas".

Once again, we thank the reviewer for noticing this. We have including main points for Trichomonas and Trypanosoma in the conclusion section.

"Lane 709 – 714: Unfitting final conclusion"

Thank you for the suggestion. The text has been deleted

"References missing , please control the text carefully

lane 51

lane 192

lane 226

lane 316, or is this still refence 90?

lane 388

lane 404

lane 427

lane 540

lane 541

lane 549

lane 632"

Thank you again for careful reading of our manuscript. Relevant references have been added in the revised manuscript.

Reviewer 3 Report

Comments and Suggestions for Authors

This manuscript discusses very interesting topic (Neutrophils antiparasitic effect). The paper is well organized, written and presented. Minor issues may be taken in consideration in the revision :

1. The title could refer to the parasites mentioned in the paper (Plsmodium, Trichomonas, Lieshmania and Entameoba) 

2. The conclusion, lines 664 to 681 should be removed (introductory part).

3. What about other parasites like Trypanosoma? 

Author Response

Reviewer 3

This manuscript discusses very interesting topic (Neutrophils antiparasitic effect). The paper is well organized, written and presented. Minor issues may be taken in consideration in the revision :

  1. The title could refer to the parasites mentioned in the paper (Plasmodium, Trichomonas, Lieshmania, Tripanosoma, and Entameoba) 
  2. The conclusion, lines 664 to 681 should be removed (introductory part).
  3. What about other parasites like Trypanosoma? 

Thank you very much for your comments and suggestions. We have corrected our manuscript accordingly.

In the title, we now refer to all parasites included in the review.

In the conclusion, we deleted the indicated text.

In our original manuscript, we did not include a section on Trypanosoma because there is very little information about neutrophils and these parasites. However, we see the relevance of this topic. Therefore, we have reviewed the manuscript by adding a new section describing Trypanosama cruzi (New section 5)
